# On the Turbulent Behavior of a Magnetically Confined Plasma Near the X-Point

**Giovanni Montani** [1,2] , **Nakia Carlevaro** [1] **and Brunello Tirozzi** [2,*]

1    ENEA, Fusion and Nuclear Safety Department, C. R. Frascati, Via E. Fermi 45, Frascati, 00044 Roma, Italy; giovanni.montani@enea.it (G.M.); nakia.carlevaro@enea.it (N.C.)
2    Physics Department, "Sapienza" University of Rome, P.le Aldo Moro 5, 00185 Roma, Italy
*    Correspondence: brunellotirozzi@gmail.com

**Abstract:** We construct a model for the turbulence near the X-point of a Tokamak device and, under suitable assumptions, we arrive to a closed equation for the electric field potential fluctuations. The analytical and numerical analysis is focused on a reduced two-dimensional formulation of the dynamics, which allows a direct mapping to the incompressible Navier-Stokes equation. The main merit of this study is to outline how the turbulence near the X-point, in correspondence to typical operation conditions of medium and large size Tokamaks, is dominated by the enstrophy cascade from large to smaller spatial scales.

**Keywords:** plasma turbulence; scrape-off-layer; Euler 2D equation

## 1. Introduction

The possibility to deal with a satisfactory confinement of the plasma in a Tokamak machine [1] is strictly related to the existence of closed magnetic surfaces [2,3] (for a discussion on the influence of dissipation effects on this feature, see Ref. [4]). However, the anomalous transport of particles and energy toward the machine walls is an intrinsic phenomenon in a Tokamak (or in medium or large size devices), and it deals with the subtle question of the power exhaust [5]. Thus, the small region of plasma between the last closed magnetic surface and the walls or the divertor (commonly dubbed Scrape-off-Layer (SoL)) plays a very critical role in present and future Tokamak experiments. Such a plasma portion has very different properties with respect to the plasma in the Tokamak core and it possesses significant collisionality, up to admit (at list for low enough frequencies) a quasi-neutral two fluid representation, in which ions and electrons are properly described in interaction, mainly via the electromagnetic force [6].

In the SoL, the magnetic field lines are always open and the presence of an X-point in the magnetic configuration, where the poloidal magnetic field identically vanishes, creates two additional "legs" in the magnetic configuration. The region just between the two legs, called "private zone", is unavoidably particularly affected by a turbulent behavior of all the fundamental (electromagnetic and thermodynamical) quantities and, over the years, it increased attention to provide a satisfactory representation for the resulting turbulent transport [7–13].

Here, we derive a local model for the turbulent dynamics of a plasma in the vicinity of an X-point, focusing attention to the basic ingredients of a predictive scenario. In addition to a constant magnetic field (taken along the toroidal $z$-direction of a Cartesian set of coordinates), we introduce the morphology of a small poloidal magnetic field, as it appears in the magnetic configuration of a region very close to the X-point and lying in the poloidal $(x, y)$ plane. The background configuration is also characterized by an equilibrium density and pressure, the former taken as an homogeneous contribution, while all the other dynamical variables live on the perturbation level only. Neglecting diamagnetic effects (ion and electron pressure gradient), the perpendicular current is then expressed by means of

the ion polarization drift velocity only. We arrive to set up a coupled dynamical system, i.e., we have to deal with a partial differential non-linear set of equations in which the unknowns (density, electric and magnetic potentials and temperature) influence each others via coupling terms.

The theoretical and numerical analysis of the turbulence profile is then restricted to the two-dimensional (ortogonal) plane $(x, y)$, which dynamical features naturally emerge as soon as the drift coupling (induced by the parallel divergence of the parallel current) is neglected. Despite this reduction, the emerging electrostatic turbulence, in the presence of ion viscosity, is then still well-individualized via a mapping with the Euler equation for a viscous incompressible fluid (for a seminal paper, see Ref. [14]). Such a correspondence concerns a direct isomorphism between the electric field potential and the so-called stream function for the fluid.

We analyze in some detail the morphology of the obtained dynamics, especially in its (truncated) Fourier representation and, for the inviscid regime, we derive an analytical solution for the (asymptotic) steady spectral morphology. This specific solution is recognized to mimic the Kolmogorov-Kraichnan enstrophy cascade spectrum proper the inertial range [15], for which the energy per unit wave-number behaves as the inverse cubed wave-number (the enstrophy flux being constant). We then utilize the so-called Arnold criterion [16] to demonstrate the stability of the analytical spectrum derived in the inviscid case, offering also more general hints on the stability in the viscous case.

The numerical analysis confirms that, if we fix the free parameters of the obtained Euler equation in the correspondence to the operation conditions of medium and large size Tokamak machines, the enstrophy cascade is dominating in the inertial range, with respect to the opposite phenomenon of an inverse cascade transferring energy from larger to smaller wave-number values. We stress that the behavior of the large scale modes resembles a phenomenon of condensation [17,18]. It is important to stress that the analytical solution is obtained by imposing an upper wave-number cut-off. In the model, we consider (as also outlined from the observed SoL turbulence phenomenology [19]) the natural cut-off coinciding with the ion Larmor radius: the typical scale of the turbulence (from millimeters to few centimeters) is larger than this characteristic ion scale (0.1 mm).

The paper is structured as follows. In Section 2, we outline the morphology properties of the magnetic configuration near the X point. In Section 3, we derive the equations describing the electromagnetic turbulence using a quasi-neutral two-fluid approach. In Section 4, we construct a local model for the turbulence dynamics near the X point in the form of a closed equation for the electric potential. The stability of the evolutionary dynamics is then discussed by means of the linear dispersion relation. In Section 5, we provide the reduced two-dimensional dynamical equation for electrostatic turbulence, outlining the analogy with the theory of incompressible flow and we study the turbulence spectral properties in terms of the vorticity dynamics. The relevant scaling of the steady energy spectrum as a function of the wave number are also analyzed. We finally study the stability of the obtained analytical spectrum specified for the inviscid case. In Section 6, the dynamics of the two dimensional electrostatic turbulence is investigated by means of a numerical code evolving the truncated Fourier expansion of the fluctuating potentials for three relevant cases. Concluding remarks follow.

## 2. Magnetic Configuration of the Equilibrium

In this Section, we fix the basic features of the local equilibrium on which we develop our turbulence analysis and the main assumptions regulating the addressed plasma scenario. We determine the morphology of the toroidal flux function, as considered sufficiently close to the X-point of a Tokamak device, while the toroidal component of the magnetic field is taken constant.

In what follows, restricting our attention to a spatial region of size much smaller than the major radius of the machine, we can neglect toroidal curvature effects and then we adopt the Cartesian coordinates $\{x, y, z\}$. The major magnetic field contribution is

along the $z$ direction, mimicking the toroidal magnetic field of a Tokamak: we denote this component by $B_0$ (here the suffix 0 denotes background quantities). Thus, the total magnetic configuration takes the following form:

$$\mathbf{B}_0 = -\partial_y \psi_0 \hat{\mathbf{e}}_x + \partial_x \psi_0 \hat{\mathbf{e}}_y + B_0 \hat{\mathbf{e}}_z \,, \tag{1}$$

where $B_0$ is taken constant and $\psi_0(x, y)$ denotes the magnetic flux function.

It is easy to realize that, if the plasma is sufficiently cold and low dense near the X-point, then the function $\psi_0$ must obey the (Ampère) equation [20]

$$\partial_x^2 \psi_0 + \partial_y^2 \psi_0 = 0 \,. \tag{2}$$

As solution of the equation above, we consider the expression $\psi_0 = (x^2 - y^2)B_{0p}/2$, where $B_{0p}$ denotes an assigned constant. This form of the magnetic flux function is depicted in Figure 1, in arbitrary units, in order to represent the X-point morphology.

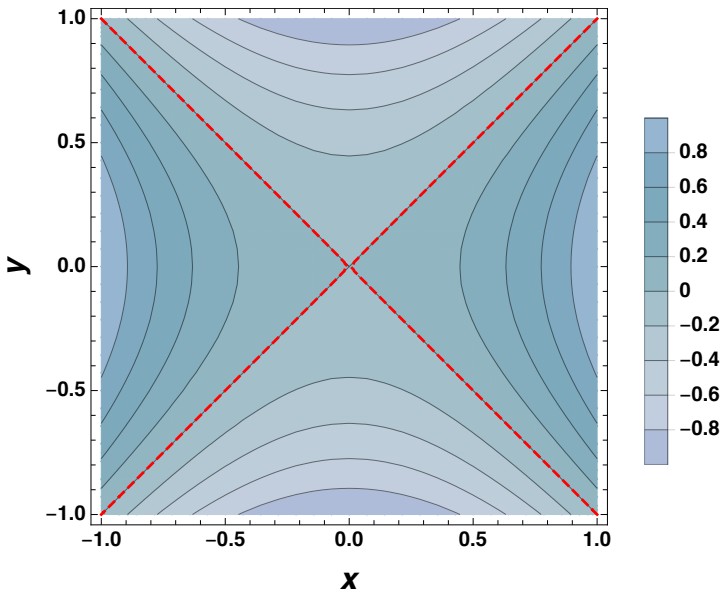

**Figure 1.** Contour plot of $\psi_0 \propto (x^2 - y^2)$ as a function of $(x, y)$ in arbitrary unit. Dashed red line represents the contour $\psi_0 = 0$.

In what follows, we will thus deal with the background magnetic field:

$$\mathbf{B}_0 = B_{0p}\left(y\hat{\mathbf{e}}_x + x\hat{\mathbf{e}}_y\right) + B_0\hat{\mathbf{e}}_z \,, \tag{3}$$

having the directional versor $\hat{\mathbf{b}}_0$. Hence, the operator $\nabla_\parallel$ takes the following expression:

$$\nabla_\parallel \equiv \frac{1}{B}\left(B_{0p}y\partial_x + B_{0p}x\partial_y + B_0\partial_z\right)\hat{\mathbf{b}}_0 \,, \tag{4}$$

where $B \equiv [B_{0p}^2(x^2 + y^2) + B_0^2]^{1/2}$.

We also deal with two other basic assumptions: (i) since the Debye length is much smaller than the turbulence scale, we assume the validity of quasi-neutrality, i.e., $n_i = n_e \equiv n$ ($n_i$ and $n_e$ denoting the ion and electron number density, respectively and we deal with a Hydrogen-like plasma); (ii) we implement the so-called "drift ordering", i.e., the smallness of the fluctuations does not prevent that their gradients are comparable (or greater) to that of the background while their second gradients dominate the dynamics. Finally, we are assuming that no velocity fields are present in the local equilibrium, so that the velocity and electric fields are pure fluctuations. Only the number density and the pressure will

contain a background contribution, here denoted by $n_0$ and $p_0$, respectively, while their fluctuation components are denoted by barred quantities, namely $\bar{n}$ and $\bar{p}$, respectively.

## 3. Dynamics of the Turbulence Fluctuations

We derive here the fundamental system of equations governing the turbulence dynamics and corresponding to a typical scheme for the non-linear low energy drift response. Our analysis is based on a two-fluid description of the plasma, in which the only relevant contribution to the ortogonal current density is provided by the ion polarization drift velocity. The starting point of the derivation is the balance law for both the parallel and the perpendicular momentum for ions and electrons, respectively. Here and in the following, we refer to parallel/perpendicular with respect to the background magnetic field versor $\hat{\boldsymbol{b}}_0$. A basic assumption, present in the model since from the very beginning, is that the plasma is characterized by a negligible parallel ion velocity.

We start by stressing that the perpendicular electron momentum conservation provides the orthogonal electron velocity as coinciding with the $\boldsymbol{E} \times \boldsymbol{B}$ velocity (we neglect the perpendicular pressure gradient, as well as other smaller effects), namely

$$\boldsymbol{v}_E \equiv \frac{cB_0}{B^2}\left(-\partial_y\phi\hat{\boldsymbol{e}}_x + \partial_x\phi\hat{\boldsymbol{e}}_y\right),\tag{5}$$

where $c$ is the speed of light (we adopt Gaussian units) and $\phi$ denotes the electric field potential fluctuation. Since, the divergence of $\boldsymbol{v}_E$ contains only first order gradients of the electric potential, we will neglect it with respect to the divergence of the ion drift polarisation velocity. Thus, from now on, we consider $\boldsymbol{v}_E$ as a divergence-less vector. Moreover, we note that, for fully fluctuating quantities (which have no background counterparts), we omit the bar notation.

The dynamics of the ion perpendicular velocity $\boldsymbol{u}_\perp$ is governed by the following equation:

$$\frac{d\boldsymbol{u}_\perp}{dt} = \frac{e}{m_i}\left(-\nabla_\perp\phi + \boldsymbol{u}_\perp \times \boldsymbol{B}/c\right) + \nu\nabla_\perp^2\boldsymbol{u}_\perp,\tag{6}$$

where $\nabla_\perp \equiv \nabla - \nabla_\parallel$ and we have neglected parallel components of the viscous stress ($e$ denotes the elementary charge, $m_i$ the ion mass and $\nu$ the kinematical (specific) ion viscosity). In the present analysis, the Lagrangian derivative reads

$$\frac{d(...)}{dt} = \partial_t(...) + \boldsymbol{v}_E \cdot \nabla_\perp(...).\tag{7}$$

Since the turbulence is, in general, observed at frequencies much smaller than the ion gyro-frequency, we can set $\boldsymbol{u}_\perp = \boldsymbol{v}_E + \boldsymbol{u}_\perp^{(1)}$ (where $\boldsymbol{u}_\perp^{(1)}$ is a small correction to the fluctuating velocity). Hence, Equation (6) gives at the leading order

$$\boldsymbol{u}_\perp^{(1)} = -\frac{c}{B_0\Omega_i}\left(\frac{d}{dt} - \nu\nabla_\perp^2\right)\nabla\phi,\tag{8}$$

where $\Omega_i$ denotes the ion gyro-frequency, calculated with the magnetic field intensity $B_0$.

If we neglect the diamagnetic effects, i.e., the pressure gradient contribution to the electron and ion velocities, the orthogonal current reads as

$$\boldsymbol{j}_\perp \equiv n_0e(\boldsymbol{u}_\perp - \boldsymbol{v}_E) = n_0e\boldsymbol{u}_\perp^{(1)},\tag{9}$$

where we also implemented the quasi-neutrality condition for the plasma and we considered $\boldsymbol{u}_\perp^{(1)}$ as the only relevant difference between the two species velocity fields. In this respect, it is worth stressing how, for the case of a constant magnetic field (as considered below), the diamagnetic velocities would be divergenceless, so that their absence has no consequences on the charge conservation, which is a basic dynamical equation in the

addressed scenario. Hence, the charge conservation equation $\nabla \cdot \boldsymbol{j} = 0$ is rewritten via Equation (8) as follows:

$$\frac{d}{dt}\nabla_\perp^2 \phi = 4\phi \frac{v_A^2}{c^2}\left(\nabla_\parallel \cdot \boldsymbol{j}_\parallel + \nu \nabla_\perp^4 \phi\right), \tag{10}$$

where $v_A$ denotes the background Alfvén velocity, calculated with $B_0$. We now account for the parallel electron momentum in the presence of a constant parallel conductivity coefficient $\sigma$ and a parallel potential vector $\boldsymbol{A}_\parallel$ only. Such a momentum balance explicitly reads

$$\boldsymbol{j}_\parallel = \sigma\left(\frac{1}{n_0 e}\nabla_\parallel p_e - \nabla_\parallel \phi - \frac{1}{c}\partial_t \boldsymbol{A}_\parallel\right), \tag{11}$$

where $p_e$ denotes the electron pressure. Now, by calculating from this equation the quantity $\nabla_\parallel \cdot \boldsymbol{j}_\parallel$, and by implementing the Lorentz gauge condition

$$\frac{1}{c}\partial_t \phi + \nabla_\parallel \cdot \boldsymbol{A}_\parallel = 0, \tag{12}$$

then, Equation (10) rewrites as

$$\frac{d}{dt}\nabla_\perp^2 \phi = \frac{4\pi}{c^2}v_A^2\left[\sigma\left(\frac{1}{n_0 e}\nabla_\parallel^2 p_e - \nabla_\parallel^2 \phi + \frac{1}{c^2}\partial_t^2 \phi\right) + \nu \nabla_\perp^4 \phi\right]. \tag{13}$$

Moreover, adopting the perfect gas law for the electron fluid and remembering that, according with the drift ordering the second gradients are dominant, we can write the expression

$$\nabla_\parallel^2 p_e = K_B T_{e0}\nabla_\parallel^2 \bar{n} + K_B n_0 \nabla_\parallel^2 \bar{T}_e, \tag{14}$$

where $K_B$ denotes the Boltzmann constant and $T_e$ the electron temperature.

To solve the electric potential dynamics, we need to couple the density and electron temperature evolution to Equation (13). The density dynamics is provided by the continuity equation (the same for ions and electrons due to the charge conservation equation, see Appendix A), i.e.,

$$\frac{d\bar{n}}{dt} + \boldsymbol{v}_E \cdot \nabla n_0 = \frac{1}{e}\nabla_\parallel \cdot \boldsymbol{j}_\parallel + \mathcal{D}_n \nabla_\perp^2 \bar{n}, \tag{15}$$

where we neglected the parallel ion velocity ($u_\parallel \simeq 0$) and the diffusion coefficient $\mathcal{D}_n$ is a phenomenological tool to model different transport regimes [12]. In this same approximation scheme, the (ideal) electron temperature evolution is governed by the following equation:

$$\frac{d\bar{T}_e}{dt} + \boldsymbol{v}_E \cdot \nabla T_{0e} = \frac{2}{3n_0 e K_B}\nabla_\parallel \cdot \boldsymbol{j}_\parallel. \tag{16}$$

In the limit in which we can neglect the ion thermal conductivity, the thermal equilibrium holds and we can speak of an equal ion and electron temperature formally both governed by the equation above.

We conclude by observing that the assumptions underlying the system of equation above, describing the X-point turbulence, are well-justified from a phenomenological point of view. Clearly, we have to think of the X-point region as that one out of the plasma separatrix, which therefore has the same qualitative morphology of the SoL. Now, the spatial scale of the turbulence runs from few millimeters to ten centimeters, a scale surely much greater than the plasma Debye length (justifying the quasi-neutrality assumption) and significantly greater than the Larmor radius of the ions (allowing to speak of low frequency dynamics). Furthermore, the mean free path of the plasma constituents is about one meters, i.e., two orders of magnitude smaller than the parallel connection length of the background magnetic field [21]. Thus, the plasma has a collisional nature and the two-fluid approximation is appropriate to describe the dynamics.

In view of these considerations, it is natural to apply, as done above, the drift ordering assumption [12,22,23]. This paradigm can be summarized by saying that the amplitude of the fluctuations is proportional to their scale and it is, in absolute value, of the order of the ratio between the electrostatic energy and the thermal one, times background values. If we denote by $(...)_1$ and $(...)_0$ the fluctuation value of a quantity and its background one, respectively (the former having a spatial scale $L_1$, while the latter $L_0$), then the drift ordering approximation can be stated as

$$\frac{(...)_1}{(...)_0} \sim \frac{L_1}{L_0} \sim \frac{e\phi}{K_B T_0} \ll 1 \,. \tag{17}$$

The relation above says that the fluctuation gradients $\sim (...)_1/L_1$ are of the same order of magnitude of the background gradients $\sim (...)_0/L_0$, while the second fluctuation derivative $\sim (...)_1/L_1^2$ clearly provides the largest contribution.

Summarizing, the crucial point in order the approximations made in constructing the dynamical system derived in this Section are valid in the SoL is that its lower temperature and higher density make the plasma therein much more collisional than the one in the core of a Tokamak.

## 4. Relevant Reduced Model

In this Section, we construct, under suitable hypotheses, a closed equation in the electric potential field only, starting from the dynamical system fixed above. The first relevant assumption we implement on the dynamics is the possibility to neglect, in Equation (15), the spatial gradients of the background density $n_0$. Then, comparing the resulting equation to Equation (10) for the electric fluctuation $\phi$, we can easily get the following relation:

$$\nabla_\perp^2 \phi = 4\pi \frac{v_A^2}{c^2} e\bar{n} \,, \tag{18}$$

where we have assumed to model the phenomenological parameter as $\mathcal{D}_n = \nu$ [12], which corresponds to the second hypothesis underling the present reduced model.

Now, focusing on a constant temperature model $T_e \equiv T_{0e} = const.$, then Equation (13) can be rewritten in the simplified closed form for the electric potential $\phi$ as

$$\frac{d}{dt}\nabla_\perp^2 \phi = \frac{4\pi\sigma v_A^2}{c^2}\partial_t^2 \phi + \frac{\sigma K_B T_{0e}}{n_0 e^2}\nabla_\parallel^2 \nabla_\perp^2 \phi - \frac{4\pi\sigma v_A^2}{c^2}\nabla_\parallel^2 \phi + \nu\nabla_\perp^4 \phi \,. \tag{19}$$

If we now introduce the dimensionless quantities $\Phi \equiv e\phi/K_B T_{0e}$, $\tau \equiv \Omega_i t$, $\bar{x} \equiv (2\pi/L)x$ and $\bar{y} \equiv (2\pi/L)y$ (here $L$ denotes a given periodicity length of the system), the equation above reads in the following dimensionless form:

$$\partial_\tau D_\perp \Phi + \alpha\left(\partial_{\bar{x}}\Phi\partial_{\bar{y}}D_\perp\Phi - \partial_{\bar{y}}\Phi\partial_{\bar{x}}D_\perp\Phi\right) =$$
$$= \gamma\left(\partial_\tau^2\Phi - D_\parallel\Phi\right) + \delta D_\perp^2\Phi + \epsilon D_\parallel D_\perp\Phi \,, \tag{20}$$

where

$$\alpha \equiv \frac{(2\pi)^2 K_B T_{e0}}{m\Omega_i^2 L^2}\,, \quad \gamma \equiv \frac{4\pi v_A^2 \sigma \Omega_i L^2}{(2\pi)^2 c^4}\,, \quad \delta \equiv \frac{(2\pi)^2 \nu}{\Omega_i L^2}\,, \quad \epsilon \equiv \frac{\Omega_i}{\nu_{ie}}\alpha\,, \tag{21}$$

here $\nu_{ie}$ denotes the ion-electron collision frequency and $D_\perp$ is the normalized orthogonal Laplace operator. Furthermore, retaining the dominant contribution in the second derivatives, we have

$$D_\parallel \equiv \frac{B_{0p}^2}{B_0^2}\left(\bar{y}^2\partial_{\bar{x}}^2 + \bar{x}^2\partial_{\bar{y}}^2 + 2\bar{x}\bar{y}\partial_{\bar{x}}\partial_{\bar{y}}\right) + (2\pi)^2\partial_{\bar{z}}^2\,, \tag{22}$$

where $\bar{z} \equiv (2\pi/L)z$. Clearly, close enough to the X-point, i.e., $\bar{x} \simeq \bar{y} \simeq 0$, we have $D_\parallel \Phi \simeq (2\pi)^2 \partial_{\bar{z}}^2 \Phi$ and $D_\perp \Phi \simeq \partial_{\bar{x}}^2 \Phi + \partial_{\bar{y}}^2 \Phi$.

*Linear Dispersion Relation*

In order to reduce the dynamics above to a coupled system of ordinary differential equation, let us now implement the Fourier representation of the field $\Phi$. Using the vector notation $\boldsymbol{\kappa} = [\boldsymbol{k}, k_z]$, with $\boldsymbol{k} = [k_x, k_y]$, for the dimensionless wave-numbers, we can write

$$\Phi = \frac{1}{(2\pi)^3} \int d^3\boldsymbol{\kappa} \, \xi_{\boldsymbol{\kappa}}(\tau) \, e^{i(k_x\bar{x} + k_y\bar{y} + k_z\bar{z})}\,, \tag{23}$$

where the integral is extended to the whole space $\{k_x, k_y, k_z\}$ and, since $\Phi$ is a real field, we have $\xi_{-\boldsymbol{\kappa}} = \xi_{\boldsymbol{\kappa}}^*$. Substituting the expansion above into Equation (20), we get, close enough to the X-point, the following equation for the Fourier component $\xi_{\boldsymbol{\kappa}}$:

$$-k^2 \partial_\tau \xi_{\boldsymbol{\kappa}} + \frac{\alpha}{(2\pi)^3} \int d^3\boldsymbol{\kappa}' \, k'^2 \big(k_x k_{\bar{y}}' - k_y k_{\bar{x}}'\big) \xi_{\boldsymbol{\kappa}-\boldsymbol{\kappa}'} \xi_{\boldsymbol{\kappa}'} =$$
$$= \gamma \partial_\tau^2 \xi_{\boldsymbol{\kappa}} + \big((\gamma + \epsilon k^2)k_z^2 + \delta \, k^4\big)\xi_{\boldsymbol{\kappa}}\,, \tag{24}$$

where we have drop the $\tau$ dependence.

Let us now investigate the behavior of the linearized system, i.e., we investigate the evolution of $\xi_{\boldsymbol{\kappa}}(\tau)$ neglecting the quadratic term regulated by the parameter $\alpha$ in Equation (24). To this end, we set $\xi_{\boldsymbol{\kappa}} \propto \exp\{-i\Omega\tau\}$, which, once substituted in the linearized equation, provides the following dispersion relation:

$$i\Omega k^2 + \gamma\Omega^2 - (\gamma + \epsilon k^2)k_z^2 - \delta \, k^4 = 0\,. \tag{25}$$

If we now separate the frequency $\Omega$ into its real part $\Omega_r$ and its imaginary one $\Gamma$, respectively (i.e., $\Omega = \Omega_r + i\Gamma$), it is easy to check that, for $\Omega_r = 0$, we get $\Gamma = (-1 \pm \sqrt{1 - 4\gamma\eta})k^2/2\gamma$, where $\eta \equiv [(\gamma + \epsilon k^2)k_z^2 + \delta k^4]/k^4$. Conversely, for $\Omega_r \neq 0$, we easily get $\Gamma = -k^2/2\gamma$ and $\Omega_r = k^2\sqrt{4\eta\gamma - 1}/2\gamma$. We clearly see that there is always a damping of the modes since $\Gamma < 0$ (no linear instability is present), but two different regimes can be distinguished: (i) when $4\gamma\eta \leqslant 1$, we deal with pure damping; (ii) when $4\gamma\eta > 1$, we have a damped oscillation of the mode. We observe that, since $\eta$ is a function of $k$ and $k_z$, the two regimes above can, in principle, co-exist but on different spatial scales.

In the case $\gamma \equiv \epsilon \equiv 0$, the linear evolution is reduced to a pure damping behavior with $\Gamma = -\delta k^2$. In this respect, we observe that, introducing the turbulent behavior of the poloidal magnetic field, via the dynamics of $A_\parallel$, we significantly affect the linear regime, simply because for small enough values of the parameter $\gamma$ (in correspondence to a fixed value of $\eta$), we can have a very small negative value of $\Gamma$, i.e., a much less damped mode appears. In the opposite case, i.e., when $\gamma$ takes sufficiently large values (a less realistic situation than the previous one), we see the emergence of weakly damped (this is a $k$-dependent feature) oscillating modes.

## 5. Two Dimensional Electrostatic Turbulence of the Reduced Scheme

Let us drop the parallel dependence in the local model developed in the previous Section. If we also set $A_\parallel = 0$ (then Equation (12) loses its applicability), Equation (20) reduces to the following form governing the two-dimensional electrostatic turbulence dynamics:

$$\partial_\tau D_\perp \Phi + \alpha \big(\partial_{\bar{x}}\Phi \partial_{\bar{y}} D_\perp \Phi - \partial_{\bar{y}}\Phi \partial_{\bar{x}} D_\perp \Phi\big) = \delta D_\perp^2 \Phi\,. \tag{26}$$

It important to stress that this equation can be mapped into the viscous two-dimensional Euler equation for an incompressible fluid. This equivalence can be easily checked via the map

$$\{v_{\bar{x}}, v_{\bar{y}}\} \to \{-\partial_{\bar{y}}\Phi, \partial_{\bar{x}}\Phi\}\,, \tag{27}$$

where $v_{\bar{x}}$ and $v_{\bar{y}}$ denote the velocity components of the incompressible flow (the viscous Euler equation is obtained taking the curl of the Navier-Stokes equation). By other words, we have a direct mapping between the electric potential $\Phi$ and the so-called "stream function" [24]. We observe that the equation of such a function would correspond to deal with $\alpha \equiv 1$ in Equation (26), which would be immediately got by choosing the length $L$ coinciding to the Larmor radius (dived by $2\pi$), or via the re-definition $\Phi \to \Phi/\alpha$.

It is a well-established result that the dynamics of an incompressible viscous fluid is associated to a turbulent behavior [15]. We thus focus particular attention to this reduced two-dimensional turbulence, in order to extrapolate well-known results established in fluid dynamics to the case of a plasma configuration very close to the X-point.

## 5.1. Inviscid Spectral Properties

Since viscosity effects on the ion dynamics are, to some extent, negligible in a Tokamak edge plasma, we rewrite here the inviscid version of Equation (26), i.e.,

$$\partial_\tau D_\perp \Phi + \partial_{\bar{x}} \Phi \partial_{\bar{y}} D_\perp \Phi - \partial_{\bar{y}} \Phi \partial_{\bar{x}} D_\perp \Phi = 0 \,, \tag{28}$$

where we intend $\Phi$ as according to $\Phi/\alpha$.

It is rather immediate to check that the equation above admits two conserved quantities, which, in fluid dynamics correspond to the specific (per unit mass) energy and the specific enstrophy, respectively. The latter explicitly reads

$$U = \int d\bar{x} d\bar{y} \, (D_\perp \Phi)^2 / 2 \,. \tag{29}$$

Let us now rewrite Equations (28) and (29) in terms of the quantity $\Pi \equiv D_\perp \Phi$ (equivalent to the vorticity of the fluid theory). We get the field equation

$$\partial_\tau \Pi + \partial_{\bar{x}} \Phi \partial_{\bar{y}} \Pi - \partial_{\bar{y}} \Phi \partial_{\bar{x}} \Pi = 0 \tag{30}$$

and the associated conserved enstrophy

$$U = \int d\bar{x} d\bar{y} \, \Pi^2 / 2 \,. \tag{31}$$

If we denote by $\Theta_k(\tau)$ the Fourier transform of $\Pi$, the representation of Equation (30) in the $k$-space reads as

$$\partial_\tau \Theta_k - \frac{1}{(2\pi)^2} \int d^2 k' \frac{(k_x k_y' - k_y k_x')}{(k - k')^2} \Theta_{k-k'} \Theta_{k'} = 0 \,, \tag{32}$$

where $\Theta_{-k} = \Theta_k^*$ and we made use of the relation $\Theta_k = -k^2 \xi_k$.

Since we are considering an isotropic plasma, the spectral representation must depend on the $k$-modulus only, i.e., we have to deal with $\Theta_k$. In order to recover statistical properties of the electric turbulence, emerging in the time averaged asymptotic state, we consider the steady spectrum $\Theta_k = \tilde{\Theta}$, where $\tilde{\Theta}$ is a complex constant, for $k \leqslant k_{max}$ and $\Theta_k = 0$ for $k > k_{max}$ (is $k_{max}$ indicates a cut-off value in the spectrum). To show that this choice of the spectrum annihilates the second term of Equation (32), we make the change of variable in the two-dimensional integral $q = k - k'$ and we adopt polar coordinates, i.e., $q \to \{\rho, \varphi\}$ (so that $k \to \{\bar{\rho}, \bar{\varphi}\}$). Thus, the integral term of Equation (32) rewrites

$$\bar{\rho} \tilde{\Theta}^2 \int_0^{\rho_{max}} d\rho \int_0^{2\pi} d\varphi (\cos \bar{\varphi} \sin \varphi - \sin \bar{\varphi} \cos \varphi) = 0 \,, \tag{33}$$

which is clearly an identity since we have introduced the physical cut-off (now implemented with $\rho_{max}$) to the spectrum, always existing in a real systems.

We can interpret the meaning of this constant spectrum for $\Pi$ by means of the mapping in Equation (27) between the fluid and plasma contexts which, in the Fourier space, links the order of magnitude of the velocity component $v_k$ in the $k$-space to $\xi_k$, according to the relation $v_k \sim k\xi_k$. In the two-dimensional fluid turbulence [15], the basic relation takes place

$$\mathcal{W}(k) \sim \mathcal{U}^{2/3} k^{-3} , \tag{34}$$

where $\mathcal{W}$ is the energy density per $k$-unit ($\mathcal{W} \sim v_k^2$ in the fluid theory) and $\mathcal{U}$ denotes the enstrophy flux. Since each $k$-mode has energy $W_k = k^2 |\xi_k|^2$, then we get

$$\mathcal{W} \sim dW_k/dk \sim k|\xi_k|^2 , \tag{35}$$

(as far as $|\xi_k|$ is a power law term in $k$) and Equation (34) yields to following relation:

$$\mathcal{U} \sim k^6 |\xi_k|^3 \sim |\Theta_k|^3 \sim |\bar{\Theta}|^3 . \tag{36}$$

As shown in Ref. [25], the studied inviscid dynamics can be associated, in the Fourier analysis, to an *ensemble* representation, which phase space is characterized by the real and imaginary parts of each Fourier component of the electric field. More specifically, it is always possible to demonstrate the equivalence of the equations in the Fourier representation to a Bose-Einstein condensate [18]. This statistical interpretation of the fluid dynamics, in terms of an ensemble, is clearly applicable also to the potential field Equation (26). Adopting the *canonical ensemble* to describe the inviscid statistical properties of the turbulence, the two fundamental constants of motion (i.e., energy and enstrophy) have to appear in the distribution function $f$ of the fluctuations, which must take the morphology $f(\xi_k) \propto \exp[-\sum(A + Bk^2)k^2|\xi_k|^2$ where $A$ and $B$ denote two inverse "temperatures", associated to energy and enstrophy, respectively [14]. The behavior corresponding to the analytic solution in Equation (36) takes place when the enstrophy constant of motion dominates the equilibrium, i.e., $k > k_c$ where $k_c$ is a critical values of the order $\mathcal{O}(A/B)$.

In the viscous case, the two dimensional turbulence tends to seek the equilibrium trough non-equilibrium states dominated by energy and enstrophy cascades. In this respect, the spectral feature described in Equation (36) has been identified by Kolmogorov-Kraichnan in the inertial range when viscosity is present [15,17,18]. Instead, our solution is exact only in the ideal case and, therefore, we can argue that this spectral shape is not significantly modified for sufficiently high Reynolds number. We recall that, formally, two kinds of inertial-transfer (cascade) regimes can take place in two dimensional turbulence: vorticity-transfer range and energy-transfer range. The analysis above mimic the first case, where indeed $\mathcal{W} \sim k^{-3}$ and an upward enstrophy cascade, from small to large mode-numbers, takes place. The second range is instead characterized by backward energy cascade, from large to small wave-numbers, and by a dependence as $\mathcal{W} \sim k^{-5/3}$ (Kolmogorov type) [15]. As we shall see below, this second shape of the spectrum is not present in our simulations of the X-point plasma according to the absence of forcing terms in the present dynamics which would guarantee the co-existence of the two distinct cascades [26,27].

Furthermore, for the inertial range, the convergence of the total enstrophy transfer rate is guaranteed by the intrinsic non-locality of in the $k$-space [28], leading to a logarithmic correction of the spectral behavior of the form

$$\mathcal{W}(k) \sim k^{-3} (\ln[k/k_i])^{-1/3} , \tag{37}$$

where $k_i$ is a typical value belonging to the bottom of the $k^{-3}$ range. By other words, we can interpret such a modification of the spectrum also as a modification of the vorticity as a function of the wave number, i.e., its steady limit would no longer be constant but behaving as $\Theta_k \sim (\ln[k/k_i])^{-1/6}$. These considerations suggest that the validity, i.e., the stability of our analytical solution, become significant for sufficiently large $k$ values.

Actually, if a cut-off in the *k*-space is considered, then the physical content of the steady spectrum constructed in Equation (36) would remain almost unaffected since the condensation phenomenon is moderate enough. It is just in the different meaning of imposing a cut-off that the fluid theory and the plasma dynamics remarkably deviate from each other. In fact, in the Navier-Stokes equation, a minimal scale for the dynamics can be reasonably inferred from the real nature of a system, but the validity of the equation is never related to the existence of such a cut-off: any small scale is, in principle, available. On the contrary, the validity of the low-frequency dynamics we proposed in this study is intrinsically valid only if the spatial scales remain sufficiently larger than the ion Larmor radius. In this respect, the Fourier truncated theory is a natural model for describing the plasma dynamics in a Tokamak edge and it makes no physical sense to take into account arbitrarily large *k*-values.

*5.2. Stability of Inviscid Flows*

Considering the above mentioned vorticity $\Pi(\bar{x}, \bar{y}, \tau)$ and stream function $\Phi(\bar{x}, \bar{y}, \tau)$, in the presence of viscosity Equation (30) can be rewritten as

$$\partial_\tau \Pi + \partial_{\bar{x}} \Phi \partial_{\bar{y}} \Pi - \partial_{\bar{y}} \Phi \partial_{\bar{x}} \Pi = \delta D_\perp^2 \Phi, \tag{38}$$

here we are assuming the analysis close enough to the X-point to get $D_\perp \simeq \partial_{\bar{x}}^2 + \partial_{\bar{y}}^2$. Viscous fluids governed by the equation above are stable because of the inequality [29]

$$\int \Pi^2 d\bar{x} d\bar{y} < -\delta \int |D_\perp \Pi|^2 d\bar{x} d\bar{y}. \tag{39}$$

This result does not hold in general but only for viscous flows in a special basin and under particular boundary conditions for the velocities.

Let us consider the non-viscous case above, i.e., $\delta = 0$ in Equation (38). We can thus use a more general result of Arnold [16] about the stability of an inviscid incompressible flow in a two dimensional domain. Suppose that $\Phi$ satisfies the following inequality

$$0 < C_1 \leqslant \frac{D_\perp \Phi}{D_\perp(D_\perp^2 \Phi)} \leqslant C_2, \tag{40}$$

in a given domain $\{\bar{x}, \bar{y}\}$, where $C_1$ and $C_2$ are given constants. Then the stream function $\Psi(\bar{x}, \bar{y}, \tau)$ associated to the generic plane wave perturbation of $\Phi(\bar{x}, \bar{y}, \tau)$ is stable:

$$\int (|D_\perp \Psi|^2 + |D_\perp^2 \Psi|^2) d\bar{x} d\bar{y} \leqslant C_3, \tag{41}$$

where $C_3$ is a positive assigned constant.

We can apply this result in our case in which the Fourier transform (dubbed $\Theta_k$) of the vorticity $\Pi$ is equal to two dimensional cylindrical function circ($k$) defined as follows

$$\Theta_k = \text{circ}(k) \equiv \begin{cases} 1 & if \quad k \leqslant 1, \\ 0 & if \quad k > 1. \end{cases} \tag{42}$$

Then the flow associated to the stream function $\Phi$ is stable in the sense above for $\bar{r} \in (\bar{r}_0, \bar{r}_1)$, which is included in the interval $(0, 1)$, where we have introduced the polar coordinates $\bar{r} = \sqrt{\bar{x}^2 + \bar{y}^2}$ and $\theta$. Let us now apply well-known properties of the Bessel function:

$$\Pi(\bar{r}) = \frac{1}{2\pi} \int_0^\infty k \, \text{circ}(k) \int_{-\pi}^\pi e^{ik\bar{r} \cos\theta} \, dk d\theta = \int_0^1 k J_0(k\bar{r}) dk = \frac{J_1(\bar{r})}{\bar{r}}, \tag{43}$$

where $J_0$ and $J_1$ are the Bessel functions of order 0 and 1, respectively, obtaining

$$D_\perp^2 \Phi = \frac{1}{r}(\partial_{\bar{r}}(\bar{r}\partial_{\bar{r}}\Phi)) = -\frac{J_1(\bar{r})}{\bar{r}}. \tag{44}$$

In this scheme, $\Phi(\bar{r})$ is then equal

$$\Phi(\bar{r}) = -J_1(\bar{r}) + \int J_0(\bar{r})d\bar{r}, \tag{45}$$

and we finally get

$$\frac{D_\perp \Phi}{D_\perp(D_\perp^2 \Phi)} = \frac{\partial_{\bar{r}}\Phi}{\partial_{\bar{r}}(-J_1(\bar{r})/\bar{r})} = \bar{r}^2 \frac{-\partial_{\bar{r}}J_1 + J_0}{J_1 - \bar{r}\partial_{\bar{r}}J_1} \equiv \tilde{g}(\bar{r}). \tag{46}$$

The function $\tilde{g}(\bar{r})$ is plotted in Figure 2, where it is evident that there is a region $\mathcal{A} : \bar{r}_0 < \bar{r} < \bar{r}_1$ in the interval $(0,1)$ where the conditions (40) are satisfied. The region is the annulus between two circles of rays $\bar{r}_0$ and $\bar{r}_1$, a case which can be adapted to a Tokamak profile. We remark that the flow generated by our choice of $\Phi$ is stationary. The theorem of Arnold applies to regions of this kind if two conditions are satisfied at the boundaries $\mathcal{A}_1 : \bar{r} = \bar{r}_0$ and $\mathcal{A}_2 : \bar{r} = \bar{r}_1$ of $\mathcal{A}$

$$\begin{cases} \oint_{\mathcal{A}_1} \partial_n \Phi = a_1, \\ \oint_{\mathcal{A}_2} \partial_n \Phi = a_2, \end{cases} \tag{47}$$

where $\partial_n$ is the directional derivative along the normal vector and the values of $a_1$, $a_2$ can be easily computed. We can interpret this result in the sense that the flow generated by a perturbation of the stationary state is stable and this correspond to our case too.

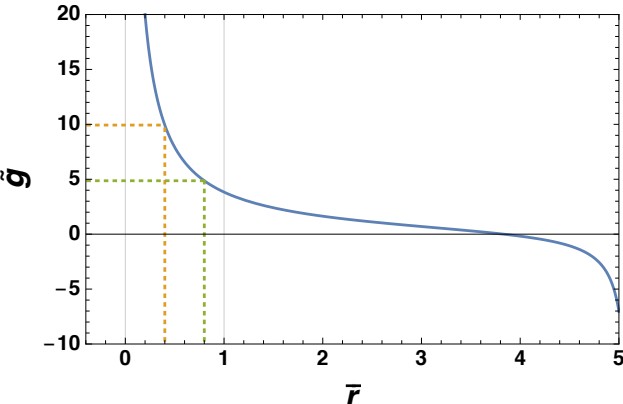

**Figure 2.** Plot of the function $\tilde{g}(\bar{r})$ of Equation (46), as a function of $\bar{r}$. The dashed lines correspond to the region $\mathcal{A} : \bar{r}_0 < \bar{r} < \bar{r}_1$.

## 6. Numerical Analysis of the Reduced Two-Dimensional Electrostatic Turbulence

The spectral properties of the two dimensional turbulence can be analyzed by means of the reduced model described in Section 4 by numerically integrating Equation (26) in the (truncated) *k*-space. In the following, we adopt the re-definition $\Phi \to \Phi/\alpha$ and we recall that $D_\perp \Phi \simeq \partial_{\bar{x}}^2 \Phi + \partial_{\bar{y}}^2 \Phi$. We also expand the electric potential fluctuation in Fourier series as follows:

$$\Phi(\tau, \bar{x}, \bar{y}) = \sum_{\ell,m} \xi_{\ell,m} e^{i(\ell \bar{x} + m\bar{y})}, \tag{48}$$

where $\ell$ and $m$ are integer (positive and negative) numbers but not both zero and reversing the sign of $\ell$ or $m$ (or both) corresponds to complex conjugation. In this scheme, Equation (26) rewrites

$$\partial_\tau \xi_{\ell,m} - \frac{S_{\ell,m}}{(\ell^2 + m^2)} + \frac{\delta}{(\ell^2 + m^2)}(\ell^4 + m^4 + 2\ell^2 m^2)\,\xi_{\ell,m} = 0\,, \tag{49}$$

$$S_{\ell,m} = \sum_{\ell',m'} (\ell'm - \ell m')((\ell - \ell')^2 + (m - m')^2)\xi_{\ell',m'}\xi_{\ell-\ell',m-m'}\,. \tag{50}$$

The numerical scheme for integrating such equations is a Runge–Kutta algorithm (4th order) evolving in time a single component $\xi_{\ell,m}$ with $(m > 0, \ell)$ or $(m = 0, \ell > 0)$. For each cycle, the reality constraint is implemented for the corresponding counterpart. The summation $S_{\ell,m}$ is technically evaluated by cycling out the components $\ell - \ell'$ (or $m - m'$) outside a chosen domain.

The energy and enstrophy introduced above (of course conserved only for inviscid fluids if $\delta = 0$) read now as

$$W = \frac{(2\pi)^2}{2} \sum_{\ell,m} W_{\ell,m}\,, \qquad W_{\ell,m} = (\ell^2 + m^2)|\xi_{\ell,m}|^2\,, \tag{51}$$

$$U = \frac{(2\pi)^2}{2} \sum_{\ell,m} U_{\ell,m}\,, \qquad U_{\ell,m} = (\ell^2 + m^2)^2|\xi_{\ell,m}|^2\,, \tag{52}$$

respectively, and we remark how the summations over $(\ell, m)$ are taken on the whole domain (in the non viscous simulations presented in this paper, they are both conserved at the order $\mathcal{O}(10^{-9})$). We also introduce the relation with the physical wave-number $K$ (in cm$^{-1}$) provided by $K^2 = (2\pi/L)^2(\ell^2 + m^2)$. In particular, for each time step, $W_{\ell,m}$ and $U_{\ell,m}$ can be evaluated by averaging over the 8 components (or 4 if $m = 0$) having the same value of $K$, thus providing the effective quantities $W_K$ and $U_K$, respectively (see, for example, the pioneering work Ref. [14]). It is important to remark the, using the analysis presented in Section 5, the mode energy $W_K$ introduced here is defined as $W_K \sim KW(K)$. By means of Equation (36) and of the relevant scaling $|\xi_K| \sim 1/K^2$, it is easy to recognize that

$$W_K \sim 1/K^2\,, \tag{53}$$

which, as already discussed, corresponds to an upward (from small to large mode-numbers) enstrophy cascade.

Since we are analyzing electrostatic turbulence in a region close to a X-point of a magnetic configuration, let us now implement realistic physical quantities specified for a typical Tokamak machine. We consider a hydrogen like plasma with $T_i = T_e = T_{0e} = 100$ eV, $B_0 = 3$ T and $n_i = n_e = n = 5 \times 10^{19}$ m$^{-3}$ [30]. The viscous parameter $\delta$ in Equation (21) is defined by means of the specific ion viscosity coefficient $\nu$ which reads [1,9]

$$\nu = \frac{1}{m_i n_i} \frac{(3/10)\, n_i K_B T_i}{\Omega_i^2 \tau_{ii}}\,, \quad \text{with} \quad \tau_{ii} = \frac{3\sqrt{n_i}(K_B T_i)^{3/2}}{4\sqrt{\pi}\, e^4 n_i \ln\Lambda_{ii}}\,, \tag{54}$$

where we set $\ln\Lambda_{ii} = 21$. Using this setup, we get the following relevant quantities: $\Omega_i \simeq 1.4 \times 10^8$ s$^{-1}$ and $\rho_i \simeq 0.05$ cm.

It is important to remark that, since we are addressing a model locally developed in a portion of the plasma near the poloidal null, we implement the physical condition of having the two dimensional periodicity box with length $L$ of the order of the centimeter. At the same time, it is well know that the physical prediction of the spectral Fourier analysis are dependent on the truncation order of the $k$-series. In this sense, since we are treating electrostatic turbulence, as previously discussed we introduce a physical cut-off at small spatial scales provided by the Larmor radius, i.e., $2\pi/K \geqslant \rho_i$.

For all the simulations, we initialize the amplitude of the electric perturbation using $e\phi \simeq 0.5$ eV (which is then scaled by the factor $\alpha$ of Equation (21)) only for the modes having 3 distinct $K$ values; all the other ones are set to zero amplitude. This corresponds to

initialize the system with 3 "rings" of modes in the squared space $\xi_{\ell,m}$ (we run a squared matrix of mode numbers $(\ell, m)$). We also underline that the energy spectrum plot presented in this Section are not instantaneous, but the relevant quantity $W_K$ is time averaged over $250\,\tau$ (the dimensionless time) in order to avoid fast statistical fluctuations. Moreover, the considered final time corresponds to the thermal equilibrium in the sense that no systematic deviations of the time averaged spectrum occur for the inviscid case.

### 6.1. Small Box (Case A)

As first case, we consider a squared box of length $L = 1.25$ cm which, using the plasma parameters defined above, yields to a viscosity coefficient $\delta \simeq 4 \times 10^{-6}$ (and to a scaling parameter $\alpha \simeq 0.06$).

We first introduce (case A1) a small scale cut-off of about two Larmor radius by considering modes with $2\pi/K \geqslant 2.5\rho_i \simeq 0.13$ cm (together with the obvious constraint $2\pi/K \leqslant L$). This yields to consider: $5 \leqslant K \leqslant 49.5$. The initial non zero modes are set as $K \simeq 7, 18, 20.6$ (cm$^{-1}$) which corresponds to run the $(\ell, m)$ mode numbers: $(1, 1)$, $(3, 2)$, $(4, 1)$ (and, of course, all the other equivalent combinations). In total, we simulate 112 modes. We first analyze the inviscid regime by setting $\delta = 0$ in Equations (49) and (50). In Figure 3, the contour plots of the fluctuations $|\xi_{\ell,m}|$ are presented for different times as function of the mode numbers. The initialization is clearly evident, while the spectral evolution indicates a more intense excitation of the small mode numbers. In Figure 4, we instead show the evolution of the modes $m = 1$ and $\ell \geqslant 0$ for a small temporal scale in order to show the saturation mechanism. The small mode number excitation is highlighted by the energy spectrum analysis presented in Figure 5 (left-hand panel), where we plot the time averaged mode energy (in the sense described above) normalized to the initial one, i.e., $W_K/W$. The chosen time corresponds to the thermal equilibrium when deviations of the spectrum no longer take place. It is evident that, using this setup, the relation (53) is properly satisfied. In the right-hand panel of Figure 5, we instead plot the spectral scaling for the viscous case by including in the simulations the parameter $\delta$. The reduction of the amount of energy in the system is evident and we observe a deviation from the scaling $W_K \sim 1/K^2$ for the first few modes.

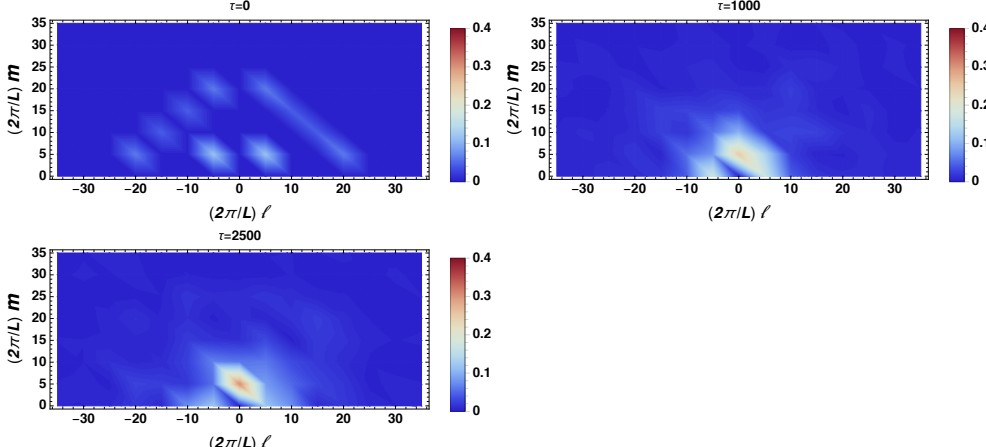

**Figure 3.** Case A1—inviscid. Contour plots of $|\xi_{\ell,m}|$ (in arbitrary units) as a function of the mode numbers scaled by $2\pi/L$. The graphs are taken at different times as indicated over the plots.

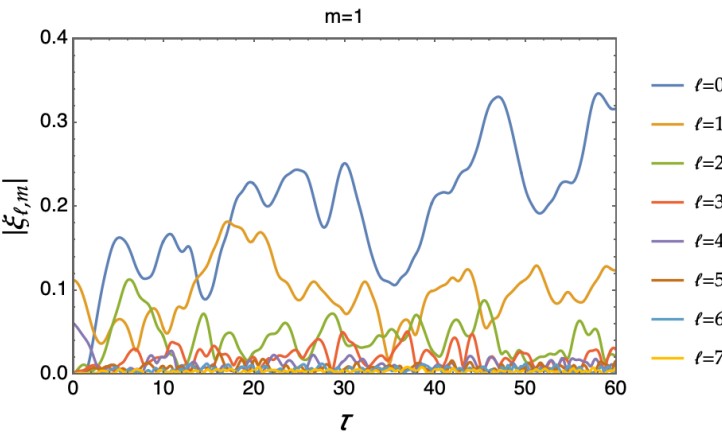

**Figure 4.** Case A1—inviscid. Early temporal evolution of $|\xi_{\ell,1}|$ with $\ell \geqslant 0$. The final time is chosen to highlight the mode saturation mechanism.

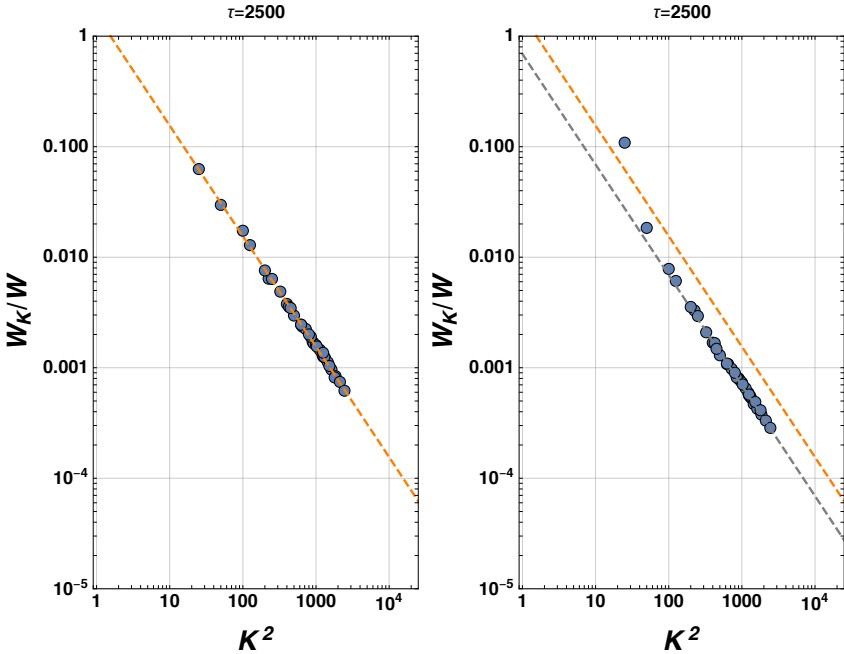

**Figure 5.** Case A1. Left-hand panel: inviscid. Right-hand panel: viscous. Log-log plots of $W_K/W$ (averaged over 250 time units) as a function of $K^2$. The energy spectra are shown at $\tau = 2500$, which corresponds to the relaxation time of the inviscid case (thermal equilibrium). The orange dashed line denotes the behavior $\sim 1/K^2$ of the non viscous regimes, while the gray dashed line is the viscous counterpart.

In this first case, the inviscid simulation in Figure 5 are thus found to properly reproduce the behavior predicted by the analytical solution discussed in the previous section. In this scenario, there is no trace of a process of constant rate of energy transport and the spectral features are essentially fixed by the vorticity dynamics. As soon as we introduce viscosity, a weak deviation from a constant vorticity spectrum is instead observed. The dissipation effect associated to a non-zero ion-ion friction in the plasma is responsible for a moderate energy trapping (a condensation process), i.e., small wave-number modes are excited to some extent, but clearly, as time goes by, the whole spectrum is progressively depressed by the viscous damping rate.

Using the same setup described above, let us now extend (case A2) the small spatial scale cut-off to the Larmor radius, i.e., considering $2\pi/K \geqslant \rho_i \simeq 0.05$ cm. This implies $5 \leqslant K \leqslant 127.3$, and, in total, we run 684 modes. The contour plots of the fluctuation evolution are depicted in Figure 6, while the early mode saturation is shown in Figure 7.

The higher excitation (with respect to case A1 in which a more stringent cut-off was present) of the low mode numbers is now evident. The energy spectrum is finally plot in Figure 8.

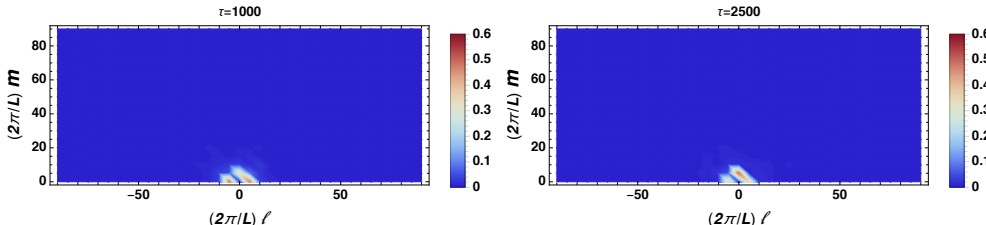

**Figure 6.** Case A2—inviscid. Contour plots of $|\xi_{\ell,m}|$ taken at distinct times as indicated over the graphs.

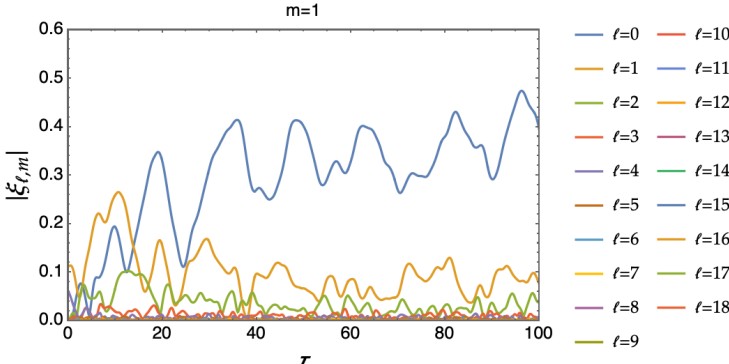

**Figure 7.** Case A2—inviscid. Evolution of $|\xi_{\ell,1}|$ with $\ell \geqslant 0$ for small temporal scales to highlight the mode saturation.

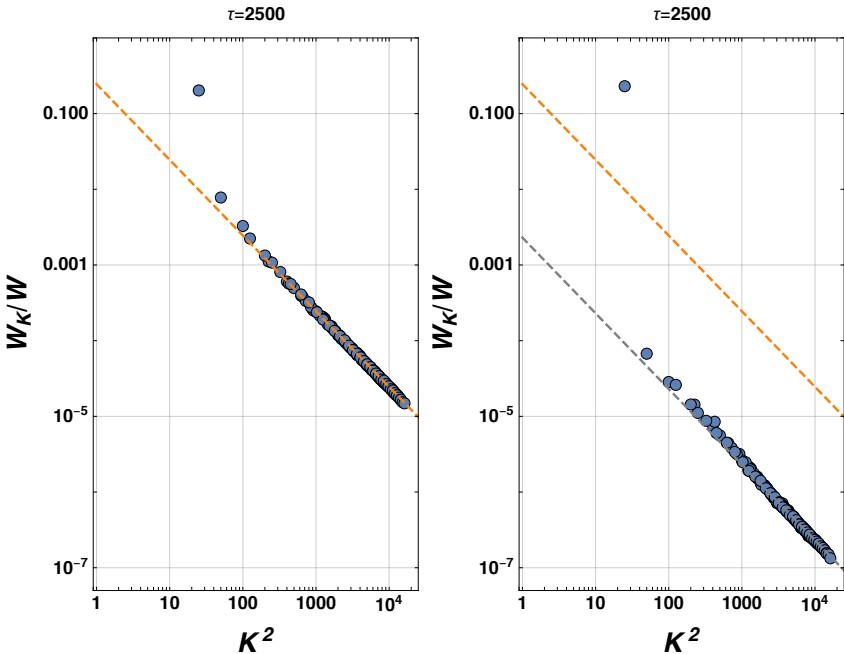

**Figure 8.** Case A2. Left-hand panel: inviscid. Right-hand panel: viscous. Plots of the averaged $W_K/W$ as a function of $K^2$, at $\tau = 2500$. As in Figure 5, the orange dashed line indicates the behavior $\sim 1/K^2$ of the non viscous regimes and the gray one corresponds to the viscous counterpart.

In particular, we can analyze the behaviour of $W_K$ for the two (non viscous) cases A1 and A2 by comparing the left-hand panels of Figures 5 and 8. A deviation from the $1/K^2$ behavior emerges when simulating a large set of modes in the case A2. In fact, having increased the maximum available values of $K$, but maintaining fixed the ratio between the

enstrophy and the energy, some modes in the low region of the wave-numbers deviate from the constant vorticity spectrum, i.e., a marked condensation phenomenon is now present according to the increase of the available $K$ value. In fact, we observe that, even if not initialized, such modes are pumped and their spectral behavior is not accounted by their analytical solution Equation (36). Moreover, this analysis confirms the relevant issue of this approach represented by the fact that the physical predictions are strongly dependent on the Fourier series truncation order. The introduction of the viscosity in the simulations (right-hand panel of Figure 8) emphasizes the shape predicted in the previous section and its stability features appear significantly confirmed by the numerical investigation in its whole set-up.

We conclude the numerical analysis of the case A2 by showing the contour plots of constant $\Phi(\bar{x}, \bar{y})$ at fixed times, constructed by inverting the Fourier transform in Equation (48). In Figure 9, we report the results at 4 distinct times for the inviscid regime. Given the form of the energy spectrum described above, the evolutive configuration of the electric potential well represents a commonplace effect of the discrete vortex theory (see the seminal paper Ref. [31]): the presence of a pair of large scale vortices mainly saturating the periodicity box.

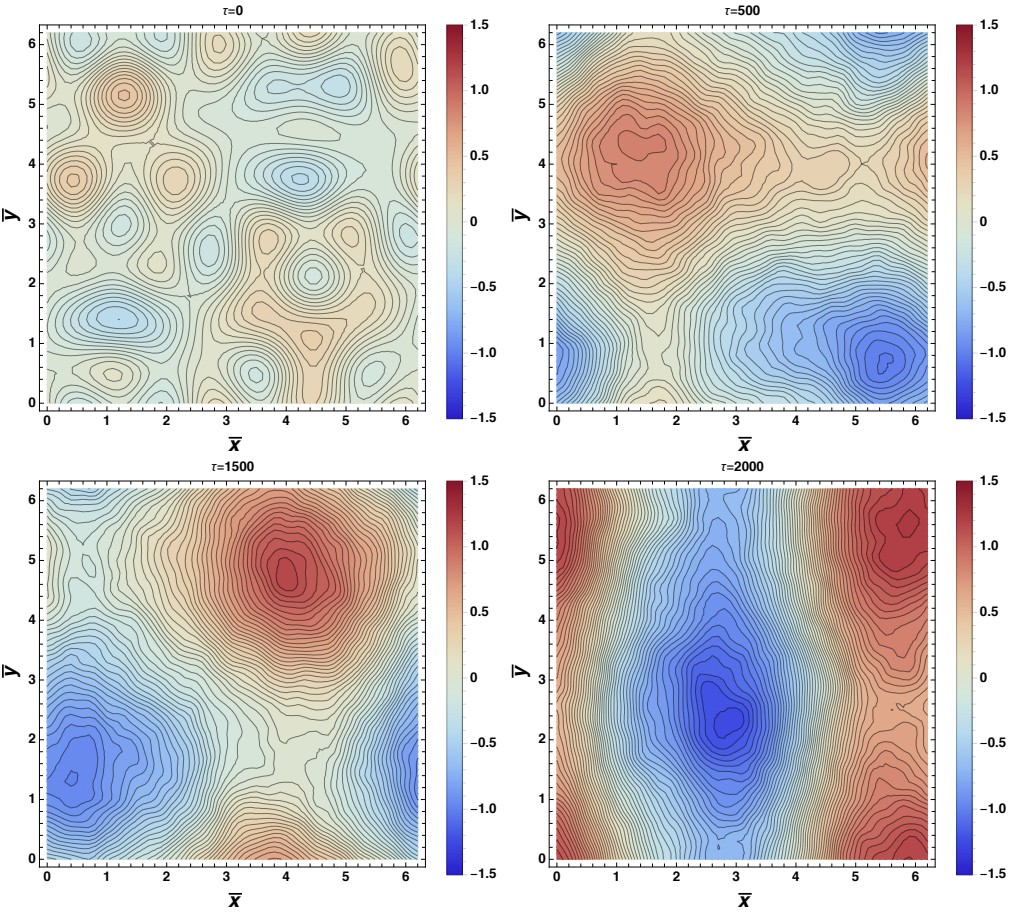

**Figure 9.** Case A2—inviscid. Contour plots of $\Phi(\bar{x}, \bar{y})$ (in arbitrary units) at fixed times as indicated over the graphs. The potential $\Phi$ is built by inverting the Fourier series.

When turning on the viscosity (see Figure 10), this feature is still present. It worth noting how the dynamics has now a negative forcing term which results in a different temporal evolution of $\Phi(\bar{x}, \bar{y})$ when compared to Figure 9. The relevant morphology related to large scale vortices is clearly outlined in the viscous case, but a time phase shift is present with respect to the inviscid regime. We also stress how viscosity yields to smooth hillsides of the $\Phi$ profile, providing a clean picture of the vortices. This is due to the fact that viscosity damps small-scale ripples.

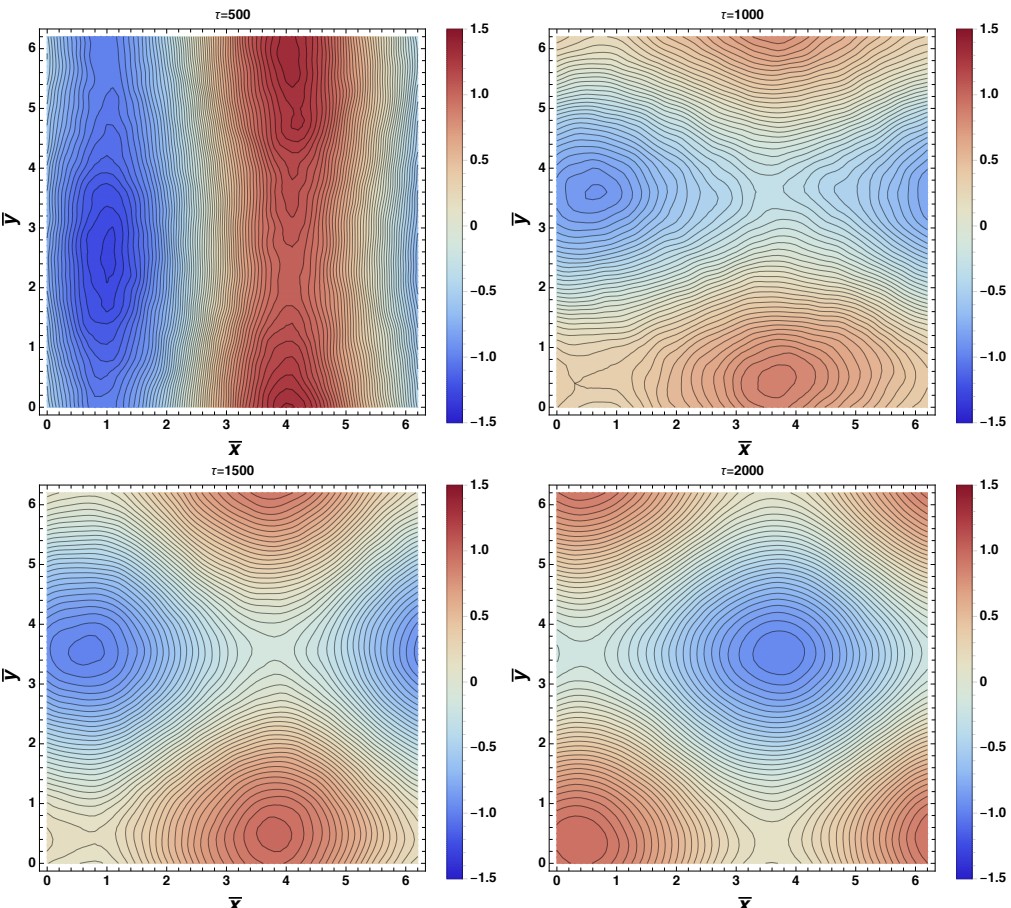

**Figure 10.** Case A2—viscous. Plots of constant $\Phi(\bar{x}, \bar{y})$ (arbitrary units) at fixed times.

### 6.2. Large Box (Case B)

For this case we now consider the squared box to have length $L = 3$ cm. Implementing the plasma parameters defined above, we get, for this case, $\delta \simeq 7.6 \times 10^{-7}$ (and the scaling parameter for the fluctuation results $\alpha \simeq 0.01$). For computational reasons, we set the cut-off $2\pi/K \geqslant 2\rho_i \simeq 0.1$ cm by considering $2 \leqslant K \leqslant 65.1$. Differently from the previous cases, non zero modes are chosen as $K \simeq 8.6$, 9.3, 11.2 (cm$^{-1}$) corresponding to the $(\ell, m)$ mode numbers: (4, 1), (4, 2), (5, 2) (plus the other combinations). In total, we thus simulate 1058 modes and we plot in Figure 11 the energy spectrum for this case (viscous and inviscid).

Such a scenario has been obtained by pushing the model parameters up to a limit case of validity for the underlying assumptions of the investigated dynamics. The relevance of such a simulation is that, due to the presence of a large number of modes, it corresponds to the more realistic situation of a nearly continuum spectrum. Here we are considering increasingly large values of $K$, and the deviations from the constant vorticity spectrum for the low $K$ portion (due to the condensation phenomenon) is again present in both inviscid and viscous cases.

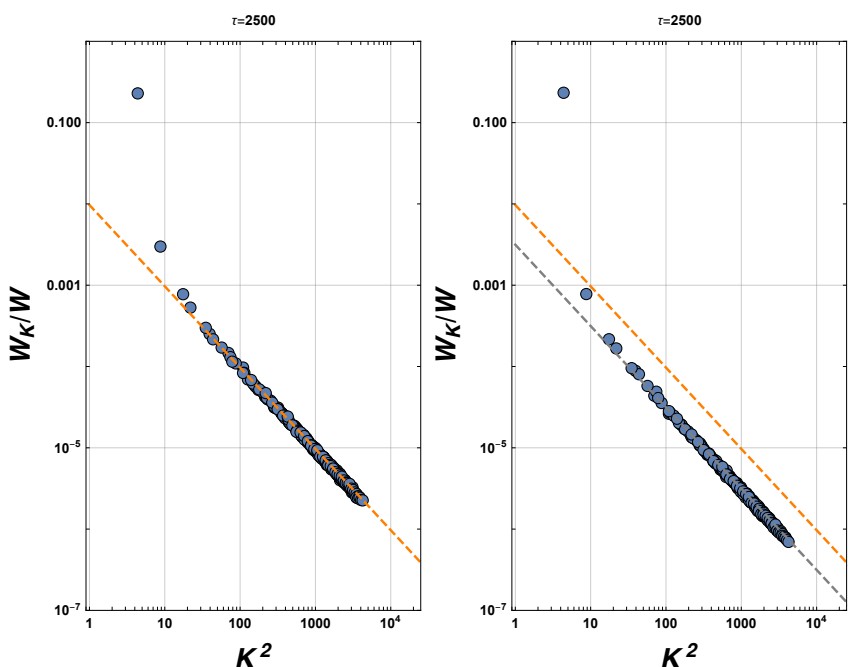

**Figure 11.** Case B. Left-hand panel: inviscid case. Right-hand panel: viscous case. Behavior of the time averaged $W_K/W$ for $\tau = 2500$ (the orange dashed line denotes the line $\sim 1/K^2$ for the non viscous regimes and the gray one is the viscous counterpart.

Furthermore, in Figure 12 we plot the spectrum compared to the theoretical $K^{-2}$ expectation (as in Figure 8, right-hand panel) and to the non-local logarithmic correction of Equation (37). We remark that, in the relation $W_K \sim K^{-2}(\ln[K/K_i])^{-1/3}$, we have considered here $K_i^2 \simeq 4$, which corresponds to the bottom of the $K$-range. It is evident from the figure that the non-local correction ensured by the logarithmic term better reproduces the simulated profile, especially for small $K$ values.

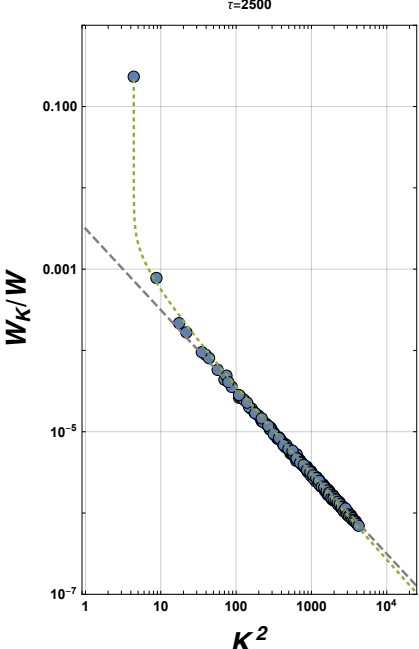

**Figure 12.** Case B - viscous. Time averaged $W_K/W$ as in the right-hand panel of Figure 11: the gray dashed line is again the behavior $\sim 1/K^2$, while the green dotted line indicates the logarithmic correction of Equation (37), i.e., the line $\sim K^{-2}(\ln[K/K_i])^{-1/3}$, with $K_i^2 \simeq 4$.

Conversely from the previous analyses, where we have shown the early mode evolution to underline the saturation mechanism, in Figure 13 we now plot the late temporal behavior of the most energetic mode $m = 1$, $\ell = 0$ in the viscous regime. The evolution points out a sort of intermittency-like feature characterized by different time intervals. This property is however not dominant and also present in the inviscid case. In fact, there are several confirmations, both experimental and numerical, that two-dimensional turbulence is not intermittent [26,32–34] or, if present [27,35], intermittency is found to be weak. Actually, large-scale intermittency is due to the developments of enhanced tails in the expected Gaussian probability density function of the perturbed field. A detailed analysis should be thus performed regarding the statistical displacements $\Delta\xi_k$, but this lies outside the scope of the present paper.

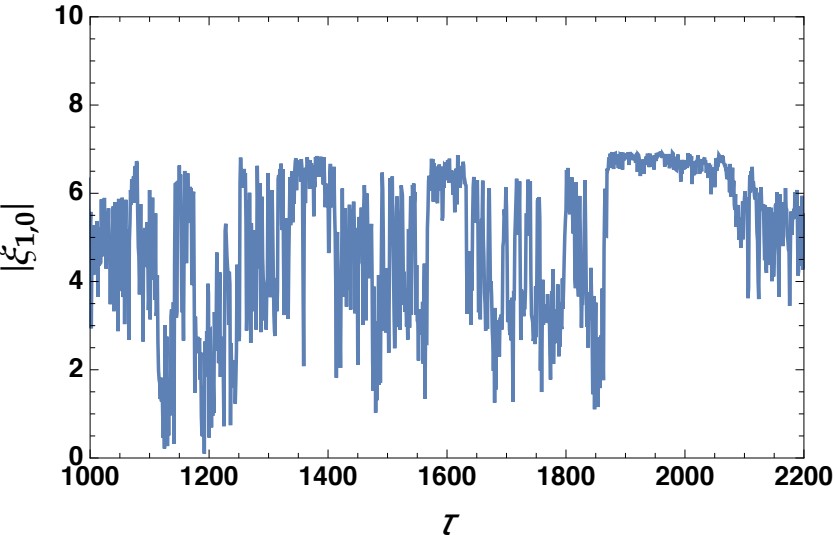

**Figure 13.** Case B-viscous. Late temporal evolution of the excited mode $|\xi_{1,0}|$. The selected time range is chosen to highlight the intermittency-like behavior.

Also for this case B, we show in Figure 14 the contour plots of $\Phi(\bar{x}, \bar{y})$ at fixed times when viscosity is turned on. The presence of the pair of rotating large scale vortices clearly emerges form the graphs, outlining again the match with respect to the discrete vortex model.

*6.3. General Remarks*

Nonetheless small deviations, we can firmly conclude that the behavior of the turbulent plasma in the operation conditions typical of the SoL of a medium or large size machine, is well represented by the analytical solution (36) (also according to previous analyses [9]). Therefore, the transport of enstrophy from small to large wave-numbers is found to be a natural feature of the Tokamak edge turbulence. According to our description of the inviscid turbulence that accounts for a Fourier representation naturally truncated by the ion Larmor radius cut-off, we can argue that the available wave-numbers are almost above the critical value $k_c \sim \sqrt{A/B}$, in the sense previously described. By other words, the SoL of a Tokamak lives in the range of parameters such that $\rho_i \ll \sqrt{B/A}$, corresponding to say that the "temperature" associated to the energy is much greater, in absolute value (negative values of $A$ are also available for the system), than the value of the "temperature" associated to the enstrophy.

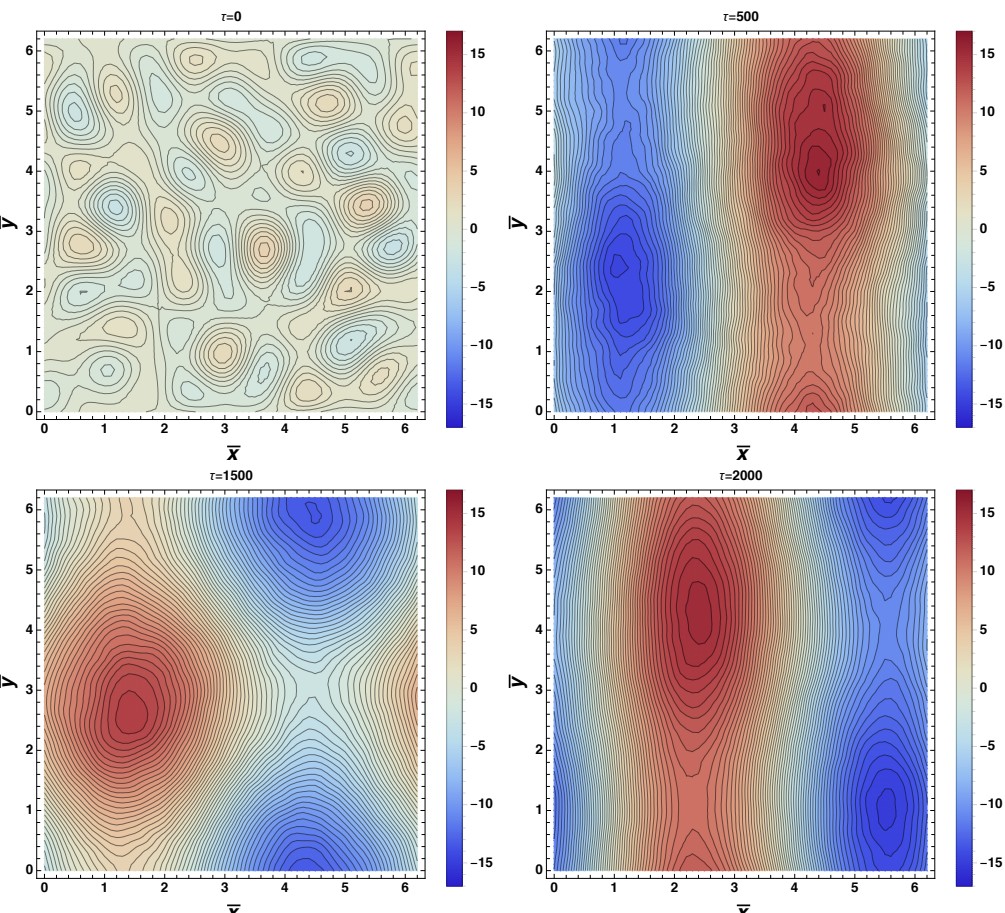

**Figure 14.** Case B—viscous. Contour plots of $\Phi(\bar{x}, \bar{y})$ at fixed times.

This situation would be clearly significantly altered in a three dimensional analysis, since the enstrophy constant of motion would be lost. However, the fact that a Tokamak has an axial symmetry structure, could imply that, at least under certain operation conditions, the three dimensional nature of the electrostatic turbulence is a weak modification of the here discussed two dimensional spectrum, so attributing a more general character to the numerical results discussed above. This claim requires a further discussion in view of the peculiarities outlined by three-dimensional turbulence with respect to the two-dimensional one [36,37]. As stated above, in the former case, the enstrophy is no longer a conserved quantity. The spectral feature is thus governed by the Kolmogorov law associated to a constant rate of energy transfer and to a scaling $\mathcal{W} \sim k^{-5/3}$. Nonetheless, we can better understand the morphology of the solution of Equation (20), by the following simple considerations. According to the topology of a Tokamak, we can naturally assume that the field $\Phi$ is periodic in $\bar{z}$ (actually, in order to reproduce a Tokamak toroidal profile, $\bar{z}$ would have to be normalized by $L' \gg L$), so that we can consider the Fourier expansion

$$\Phi(\tau, \bar{x}, \bar{y}, \bar{z}) = \sum_n \chi_n(\tau, \bar{x}, \bar{y}) e^{in\bar{z}}. \tag{55}$$

Substituting the expansion above into Equation (20), we get the following dynamics for the Fourier harmonics $\chi_n$:

$$\partial_\tau D_\perp \chi_n + \alpha \sum_{n'} \left( \partial_{\bar{x}} \chi_{n-n'} \partial_{\bar{y}} D_\perp \chi_{n'} - \partial_{\bar{y}} \chi_{n-n'} \partial_{\bar{x}} D_\perp \chi_{n'} \right) =$$
$$= \gamma (\partial_\tau^2 + n^2) \chi_n - \delta D_\perp^2 \chi_n + \epsilon n^2 D_\perp \chi_n. \tag{56}$$

To this equation we must add the gauge condition for each harmonic $A_{||n}$ of the parallel magnetic potential vector.

Despite in constructing Equation (20) we neglected the number density gradients, it is a well-known result that such a term is responsible for the linear drift wave instability. The idea proposed in Ref. [9] is that, in the left-hand side of Equation (56) the $n = 0$ mode dominates. At the same time, in the right-hand side, the value of $n$ maximizing the linear growth rate is associated to this mode. In the spirit of this postulation we can thus expect that the three-dimensional turbulence spectrum remains close the one discussed above. However, since it is well-known [10] that the non-linear drift response occurs when the linear growth rate is small enough, the investigation of the fully developed three-dimensional turbulence, as well as its deviation from the $k^{-3}$ law, constitute a valuable field for future investigations.

We also note that, in a real Tokamak device setting, the background magnetic field lines have non-zero curvature and this feature has a relevant impact on the turbulence morphology. In this respect, see the analysis developed in Ref. [11] where the interaction between ballooning mode and the non-linear drift response is discussed.

### 7. Conclusions

We analyzed the basic dynamical scheme regulating the turbulent behavior of the plasma in a Tokamak SoL. We focused on low frequency phenomenology (i.e., the typical frequency of the fluctuations is smaller than the ion gyro-frequency) and on a local model nearby the X-point, so that the magnetic configuration of the equilibrium has been modelled via a dominant contribution along the $z$-axis and a smaller poloidal contribution, exactly vanishing in the X-point taken as origin of the $(x, y)$ plane.

After the general set up of the problem, the theoretical and numerical analysis has been devote to the simplified two-dimensional electrostatic turbulence, also in the presence of ion viscosity. This formulation was investigated using the natural mapping existing with a two-dimensional turbulent incompressible fluid: the electric potential field and the fluid stream function are described by same dynamics.

The inviscid turbulence was discussed on a theoretical framework by outlining the existence of a steady analytical solution for the (truncated) spectral representation, which corresponds to a $\sim K^{-3}$ behavior. Then, the stability of this solution has been properly investigated and, using the Arnold criterion, we arrive to conclude that such a spectral profile is a stable configuration for the system evolution.

The numerical analysis has confirmed that, in the range of parameters available in the SoL of a medium or large size Tokamak, the enstrophy cascade toward larger wave-numbers dominates the asymptotic spectrum both in the inviscid and viscous cases. However, a certain energy trapping into the larger spatial scales is observed, especially when the viscosity contribution is considered.

Concluding, we stress how having fixed all the basic features of the turbulence due to the advective transport of the "vorticity" offers a dynamical and physical scenario in which the role of the non-linear drift response can be properly evaluated [11].

**Author Contributions:** Conceptualization, G.M.; methodology, G.M. and N.C.; software, N.C.; validation, N.C.; formal analysis, B.T.; investigation, B.T., G.M. and N.C.; resources, G.M., N.C. and B.T.; data curation, N.C.; writing—original draft preparation, G.M., N.C. and B.T.; visualization, N.C. All authors have read and agreed to the published version of the manuscript.

**Funding:** This research received no external funding.

**Institutional Review Board Statement:** Not applicable.

**Informed Consent Statement:** Not applicable.

**Data Availability Statement:** No data repository is available.

**Acknowledgments:** We want to thank Francesco Cianfrani for the interesting discussion on this subject. NC wants to thank Giulio Rubino for his comments on the numerical code.

**Conflicts of Interest:** The authors declare no conflict of interest.

**Appendix A**

Our analysis is based on the assumption of quasi-neutrality $n_i \equiv n_e$ and with the following requests on the two specie velocity fields:

$$\boldsymbol{v}_e = \boldsymbol{v}_E + \boldsymbol{v}_\parallel \,, \tag{A1}$$

$$\boldsymbol{u}_i = \boldsymbol{v}_E + \boldsymbol{u}_\perp^{(1)} \,. \tag{A2}$$

It is worth noting that, if the magnetic field is uniform (as *de facto* taken in this work), then $\nabla \cdot \boldsymbol{v}_E = 0$, while the velocity $\boldsymbol{u}_\perp^{(1)}$ survives in the equations only under the divergence operator. The electron continuity equation reads as:

$$\partial_t n_e + \boldsymbol{v}_E \cdot \nabla_\perp n_e + \nabla_\parallel \cdot (n_e \boldsymbol{v}_\parallel) = 0 \,, \tag{A3}$$

which, neglecting the ion parallel velocity, can be easily restated in the form:

$$\partial_t n_e + \boldsymbol{v}_E \cdot \nabla_\perp n_e = \frac{1}{e} \nabla_\parallel \cdot \boldsymbol{j}_\parallel \,. \tag{A4}$$

The ion continuity equation takes the explicit form:

$$\partial_t n_i + \boldsymbol{v}_E \cdot \nabla_\perp n_i + \nabla_\perp \cdot n_i u_\perp^{(1)} = 0 \,. \tag{A5}$$

Since the quasi-neutrality requires $n_i \equiv n_e$, the compatibility between Equations (A4) and (A5) is provided by the relation

$$\nabla_\perp \cdot (e n_e \boldsymbol{u}_\perp^{(1)}) = -\nabla_\parallel \cdot \boldsymbol{j}_\parallel \,, \tag{A6}$$

which is noting more than the charge conservation law.

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
