# Peer review of "On the Turbulent Behavior of a Magnetically Confined Plasma near the X-Point"

_fluids, doi:10.3390/fluids7050157_

Round 1

Reviewer 1 Report

Summary of the Work

The aim of this work is to study the turbulent behaviour of the plasma in a Tokamak Scrape-off-Layer (SOL) by proposing an ad hoc local model nearby the X-point in the form of a closed equation for the electric potential. The equilibrium magnetic configuration corresponds to a dominant contribution along the z-axis and a smaller poloidal contribution, vanishing in the X-point. The authors investigated the two-dimensional electrostatic turbulence in presence of ion viscosity. The typical frequency of the fluctuations is supposed to be smaller than the ion gyro-frequency. The authors analysed the scaling of the steady energy spectrum as a function of the wave number and they studied the stability of the analytical spectrum obtained for the inviscid case. The analytical solution describing the Kraichnan enstrophy cascade is obtained by imposing an upper wave-number cut-off. Both the inviscid and viscous spectrum have been analysed by a numerical code for values of the physical parameters of a plasma near the X-point of medium or large size Tokamak devices.

Main Results Obtained

- The behaviour of the turbulent plasma in the operation conditions typical of the SOL of a medium or large size machine is well provided by the analytical solution (ref. to Eq. (33) on page 8 of the manuscript).

- Transport of enstrophy from small to large wave-numbers is a characteristic of the Tokamak edge turbulence.

General Remarks

- The work is clearly written and I enjoyed to read it.

- However, the authors have often failed to define some notations and, at times, the limits of validity of some adopted assumptions. The definitions and assumptions to which I refer are well known to the community of fusionists but, for easy reference, it is customary to provide these definitions as they appear for the first time in the manuscript.

- Several results obtained in this work were already known in the literature (in particular in the field of fluid theory).

- The list references must be completed; other relevant works on the cascade of enstrophy in two-dimensional turbulence have appeared in the literature.

- Some important points need further clarification.

The following suggestions are intended to fill some gaps and clarify some points.

Suggestions

Let us start with the definitions.

1) Even though it is well known in the Fusion literature, for the sake of clarity, please define the ion perpendicular velocity u ("perpendicular" to the magnetic line).

2) Please, define the operator ∇≡.

Clarification about one of the main assumptions.

3) Turbulent behaviour has been analysed by adopting the drift ordering. This assumption has been used to get Eq. (13), necessary to solve the electric potential dynamics (12) (ref. to page 5 of the manuscript). The authors justified this assumption by stating that “the smallness of the fluctuations does not prevent that their gradients are comparable (or greater) to that of the background while their second gradients dominate the dynamics”. The authors are asked to clarify this assumption from a physical point of view - by mentioning its limit of validity - in case of turbulent Tokamak-plasmas near the X-point. By other words, why, near the X-point, are the fluctuations so small that they don't prevent the second gradients from dominating the dynamics? We may object that this assumption is fully justified to get the fluid equations for plasmas in a collisional regime (ref., for instance, to A. N. Simakov and P. J. Catto, PSFC/JA-03-13), but its validity is compromised when Tokamak-plasmas is in turbulent regime (ref., for instance, to R. Balescu book).

Energy spectrum in two-dimensional turbulence.

4) The key result obtained in this work is the analytical solution (33) (see page 8 of the manuscript). For clarity, please explain the presence of a multiplicative "k" in the relation W(k)∼kk|2 (this is the most important relation as the other ones derive trivially from this one when combined with the Kraichnan et al. cascades (ref. to Eq. 32)).

5) The Kraichnan et al. energy spectrum reads W(k)=cU2/3k−3 where U is the enstrophy dissipation and “c” is of the order of unity. However, in reality, “c” is varying between different simulations and is not a perfect constant. This variation is interpreted as a consequence of large-scale dissipation intermittency. In performing numerical analysis of two-dimensional electrostatic turbulence for a large box (case B), have the authors observed this phenomenon?

6) The final version of the Kraichnan et al. energy spectrum contains a logarithmic correction W(k)= cU2/3k -3 [In (k/k1)]-1/3 where k1 is a wavenumber at the bottom of the range. The inclusion of this term is very important as this correction ensures the convergence of the integrations providing the rates at which energy and enstrophy are transferred (from wave-numbers < k to wave-numbers > k, respectively). The authors are asked to comment very briefly the relevance of the logarithmic term in relation of the final conclusion of their study establishing that the behaviour of the turbulent plasma in the SOL of a medium or large size machine is well represented by the analytical solution (33). Notice that, as said, the analytical solution (33) does not ensure the convergence of the above-mentioned integrals.

7) This question is somewhat related to the previous one. In their approach, the authors introduced a physical cut-off (i.e., a kmax corresponding to a cut-off value in the spectrum). The authors justified the presence of a cut-off value as “it always exists in real systems”. This aspect has been extensively discussed in the past literature. Indeed, according to closure approximations, in two dimensions the equilibrium state for kmax=∞ corresponds to one of condensation of all the energy in the gravest modes. Moreover, consider that the energy spectrum, with the supplementary logarithmic contribution, carries enstrophy upward and extends to successively higher “k” as time goes on. So, we may object that if we suppose that the cut-off value does not exist, but we take into consideration the logarithmic contribution of the energy spectrum of Kraichnan et al., the presence of this term may affect the analysis performed by the authors for Tokamak-plasmas near the X-point significantly. The authors are asked to provide their comment, not only qualitatively, about this issue.

Remark

Since Tokamak has an axially symmetrical structure, the authors stated that, under certain operating conditions, the three-dimensional nature of electrostatic turbulence cannot but bring a slight modification to the results obtained by analysing the two-dimensional spectrum. I would like to draw the attention of the authors on the following. Kolmogorov’s theory of the inertial range has a central place in the theory of three-dimensional turbulence. In this regard:

i) The energy spectrum in three-dimensional case reads W(k)= cU2/3 k-5/3 where the presence of the exponent -5/3 takes into account the intermittency effects. This effect has not been quoted in the present work;

ii) For three-dimensional turbulence it is commonly accepted that the inertial range can develop to infinite wave-numbers in finite time if the viscosity tends to vanish;

iii) In two-dimensional turbulence there are two conserved quantities: energy and enstrophy. However, enstrophy is not preserved by the fully 3D inviscid dynamics (S. Musacchio et al. Physics of Fluids 2017).

Finally, there are several works appearing in the literature confirming that the enstrophy spectrum is a robust feature of the two-dimensional fluid equations. However, there is a complete lack of universality of higher-order statistics of vorticity increments in the enstrophy cascade-range.

In addition to the points mentioned above, I also add that

iv) If on the one hand it is correct to say that in Tokamak-plasmas the dynamics is axially symmetric, which reduces the number of spatial variables to two (r, θ), on the other hand we have to take into account that plasmas are magnetically confined in a toroidal configuration, having, therefore, a global geometry endowed with curvature, and not confined in a "flat" Cartesian geometry. As the authors know, the presence of a curvature has a great impact on the dynamics of Tokamak-plasmas, in particular in terms of radial losses (energy and matter Tokamak-losses).

Conclusions

The work is certainly interesting and, in my opinion, deserves to be published. However, there are several points that need to be clarified. Written in this form, it could lead the reader to conclude (albeit hastily) that basically the work does not bring any innovative results other than to confirm that near the X-point the energy spectrum turns out to be that of Kraichnan et al. (by virtue of the analogy with a two-dimensional Eulerian fluid without, however, taking into account the contributions of Frisch, Rose, etc.). The authors are therefore invited to take into account the above suggestions.

Author Response

The answers to referee1 have been uploaded as a pdf file

Reviewer 2 Report

In the current manuscript, the authors studied some turbulence behaviors of a plasma near the X-point under magnetically confined conditions, including the magnetic configuration properties, derived equations, et al. The topic is interesting, while some key comments are listed below:

  1. The title of the manuscript is “On the turbulent behavior of a magnetically confined plasma near the X-point”. Here we can see three keywords, i.e., turbulent behaviors, magnetically confined plasma, and X-point. While have the authors focused on these key items in the current manuscript? From the Abstract the answer is NO.
  2. Please consider rewriting the Abstract. All contents in current version just show how the authors conducted the research. Only the last sentence showed the results. Please keep in mind this is research article rather than an experimental report in the class. Please used no more than two sentences showed how you conducted your research, then your results, then your findings, then your discussions and implying practical applications. BTW, why does the authors like use “limited”, please count how many limit/limited(s) used in the Abstract and manuscript! The authors should learn more how to write a scientific paper.
  3. Please consider rewriting the conclusions. Also use no more than two sentences to summarize how the authors conduct the research, the conclude what have been discovered. The current version still belongs to discussion part.
  4. Cannot understand what the authors wanted to express in Line 42-43. We consider as negligible the ion and electron diamagnetic velocity and the perpendicular current is then due to the ion polarization drift velocity only. English writing skills need to be improved.
  5. What is the meaning a coupled dynamical system? Which parameter is coupled which coupled parameter? Please state clear in Introduction.
  6. Compare Figures 9 and 10, the Φ is so different between inviscid and viscous. How? We did not observe the explanations.
  7. So many typo issues, i.e., the parenthesis in Figure 11.

Author Response

The answer to referee 2 are submitted in the pdf file

Round 2

Reviewer 1 Report

The previous version of the work showed some vulnerable parts. The authors have filled the gaps by answered satisfactorily to the series of questions raised in my previous report and adding supplementary comments to the present version of the manuscript. I confirm that, for me,  the work is interesting and deserves to be published.

Reviewer 2 Report

The revised manuscript has been improved a lot. While the typo and grammar issues still have not been addressed throughout the manuscript here and there. Also, the revised Abstract is terrible. Please ask the help from a professional to address these issues before it can be accepted to publish. Here I only pointed out the issues in the first two sentences listed as below:

We construct a model for the turbulence (constructed a turbulent model) near the X-point of a Tokamak device (cannot understand what the authors wanted to express here) and, under suitable assumptions, we arrive (proposed) to a closed equation for the electric field potential fluctuations (please check one sentence or two sentences here???). The analytical and numerical analysis is focused (focused ) on a reduced two-dimensional formulation of the dynamics (cannot understand what the authors wanted to express here),  which allows a direct mapping to the incompressible Navier-Stokes equation.